# Recent Advances of Ti/Zr-Substituted Polyoxometalates: From Structural Diversity to Functional Applications

**DOI:** 10.3390/molecules27248799

**Published:** 2022-12-12

**Authors:** Zhihui Ni, Hongjin Lv, Guoyu Yang

**Affiliations:** 1Center for Advanced Materials Research, Zhongyuan University of Technology, Zhengzhou 450007, China; 2MOE Key Laboratory of Cluster Science, School of Chemistry and Chemical Engineering, Beijing Institute of Technology, Beijing 102488, China

**Keywords:** polyoxometalates, titanium, zirconium, transition metal substitution

## Abstract

Polyoxometalates (POMs), a large family of anionic polynuclear metal–oxo clusters, have received considerable research attention due to their structural versatility and diverse physicochemical properties. Lacunary POMs are key building blocks for the syntheses of functional POMs due to their highly active multidentate O-donor sites. In this review, we have addressed the structural diversities of Ti/Zr-substituted POMs based on the polymerization number of POM building blocks and the number of Ti and Zr centers. The synthetic strategies and relevant catalytic applications of some representative Ti/Zr-substituted POMs have been discussed in detail. Finally, the outlook on the future development of this area is also prospected.

## 1. Introduction

Polyoxometalates (POMs), as anionic metal-oxide clusters with diverse nuclearities, elemental compositions and physicochemical properties, have usually been constructed through the self-assembly of reactive oxometallate precursors in aqueous or organic reaction systems. [1,2,3,4] POMs can serve as crucial intermediates in the reaction pathway from water-soluble metal ions to insoluble metal oxides, and isolation of these molecular intermediates enable insightful elucidation on the formation mechanism and control over reaction pathways. POMs exhibit special characteristics of high negative charges, rich redox properties, good thermal stability, and readily available organic grafting [5,6], leading to wide applications in catalysis [7], magnetism [8], material science [9], electrochemistry [10], luminescence [11], etc.

As an important derivative of plenary POMs, lacunary POMs can be easily formed by removing one to several [MO_6_] (M = Mo, W) building blocks from prototypal architectures such as the Keggin or Wells–Dawson type POMs [12]. These lacunary POMs usually show high coordination reactivity and oxidative and thermal stability. Their high negative charge and nucleophilic oxygen-enriched surfaces render them suitable inorganic, diamagnetic, multidentate nucleophilic ligands toward the electrophilic center. Transition metals (TM) or lanthanide (Ln) cations can be easily incorporated into the defect sites of lacunary POM ligands to construct metal-substituted POMs, which can exhibit unique physicochemical properties depending on the types of incorporated metal ions [13,14,15,16,17,18,19,20]. Metal-substituted POMs (MSPs) typically possess a higher negative charge density than that of the plenary POMs due to the substitution of a high oxidation state M^6+^ ion (e.g., W^6+^, Mo^6+^) with a low oxidation state M^n+^ ion (usually *n* = 1–3) [21]. To date, a wide variety of MSPs have been prepared, especially by the transition metals like manganese, iron, cobalt, nickel, copper and zinc in the fourth period and the lanthanides in the sixth period of the periodic table [22,23]. In contrast, the research on the syntheses of titanium- and zirconium-substituted POMs is still in a very early stage, which could be mainly attributed to the following two reasons: (a) the easy hydrolysis of Ti^4+^/Zr^4+^ salts in aqueous synthesis, and (b) the high tendency to formation of isolate oligomeric structures through intermolecular dehydration of terminal hydroxyls.

In this review, we have mainly focused on the structural diversities of Ti/Zr-substituted POMs according to the polymerization number of POM building blocks and the number of titanium or zirconium atoms. The representative catalytic application of Ti/Zr-substituted POMs has also been discussed. Finally, a perspective of this research area is also proposed. It is expected that this review could provide research insights into the controllable design and syntheses of Ti/Zr-substituted POMs derivatives with interesting catalytic properties.

## 2. The Syntheses and Structures of Ti/Zr-substituted POMs

### 2.1. Ti/Zr-Substituted Monomeric POMs

It is well known that titanium/zirconium-based compounds (e.g., TiO_2_, ZrO_2_) have been widely used in the fields of energy conversion, catalysis, and environmental treatment. As interesting molecular models of TiO_2_ and ZrO_2_ structures, the syntheses of titanium-/zirconium-substituted POMs could be dated back to the 1980s. In 1983, Knoth et al. reported the first case of Ti-substituted Keggin-type monomeric [TiW_11_PO_40_]^5−^ polyoxoanion cluster (Figure 1a), which was prepared by the reaction of monovacant (Bu_4_N)_4_H_3_W_11_PO_39_ with titanium tetrachloride in dichloroethane solution [24]. In 2000, Kholdeeva and co-workers also reported two similar cases of [PTiW_11_O_40_]^5−^ and [PTiW_11_O_41_]^7−^ polyoxoanions [25]. Qu et al. reported Ti-substituted Dawson-type monomeric α_2_-[P_2_W_17_(TiO_2_)O_61_]^8−^ polyoxoanion (Figure 1b), which was synthesized from vacant heteropolytungstate precursors α_2_-[P_2_W_17_O_61_]^10−^ and Ti(SO_4_)_2_ using an aqueous solution-based synthetic approach [26]. Successively, Ti_3_-substituted monomeric POM has also been reported with multi-lacunary POMs α-1,2,3-[P_2_W_15_O_56_]^12−^ as precursors. For instance, Nomiya et. al. successfully reported a tris-[peroxotitanium(IV)]-substituted α-Dawson monomeric [*α*-1,2,3-P_2_W_15_(TiO_2_)_3_O_56_(OH)_3_]^9−^ polyoxoanion (**1,** Figure 1c), which are derived from {[*α*-1,2,3-P_2_W_15_Ti_3_O_59_(OH)_3_]_4_[*µ*_3_-Ti(OH)_3_]_4_Cl}^33−^ (**2a**) in 30% aqueous hydrogen peroxide solution. Thereinto, the four bridging Ti octahedral groups of **2a** were considered as the crucial roles for the synthesis of polyoxoanion **1 [27]**. Subsequently, they prepared [[{Ti(OH)(ox)}_2_(*μ*-O)](*α*-PW_11_O_39_)]^7−^ (Figure 1d) by using the tri-lacunary species of [A-PW_9_O_34_]^9−^ and the anionic titanium(IV) complex as precursors with the molar ratio of 1:2 under acidic conditions. The molecular structure can be recognized as a hybrid containing one mono-lacunary POM ligand and two octahedral Ti-oxo moieties [28]. Then, a tetra-Ti-substituted di-lacunary α-Keggin monomeric [[{Ti(ox)(H_2_O)}_4_(*µ*-O)_3_](*α*-PW_10_O_37_)]^7−^ polyoxoanion (Figure 1e) was also prepared, which was constructed by using the dimeric dititanium(IV)-substituted POM [(*α*-1,2-PW_10_Ti_2_O_39_)_2_]^10−^ as precursor under acidic conditions [29]. Additionally, [[{Ti(H_2_O)_3_}_2_{Ti(H_2_O)_2_}_2_(*μ*-O)_3_(SO_4_)](PW_10_O_37_)]^−^ polyoxoanion (Figure 1f) was synthesized through the reaction of Ti(SO_4_)_2_ with [(*α*-1,2-PW_10_Ti_2_O_38_)_2_O_2_]^10−^ and [(*α*-1,2,3-PW_9_Ti_3_O_37_)_2_O_3_]^12−^ under strongly acidic conditions, where the tetra-titanium(IV) oxide cluster was anchored onto the binding sites of lacunary Keggin POM [30]. In 2018, An et al. reported two organic-inorganic hybrid POMs, [(H_2_O)_4_(3-Hpic)_2_Ln][(H_2_O)_5_(3-Hpic)_2_Ln][PW_10_Ti_2_O_40_]^−^ (Ln= Ce, Nd, Sm), where binuclear Ti-substituted Keggin-type [PW_10_Ti_2_O_40_]^7−^ polyoxoanion was further modified by four Ln-3-Hpic coordinating groups [31]. Recently, Poblet et al. successfully prepared the other organic-inorganic hybrid POM, [B-*α*-SbW_9_O_33_(*^t^*BuSiO)_3_Ti(O^i^Pr)]^3−^, by anchoring Ti(O^i^Pr) moiety on the silanol-functionalized an antimony-containing trilacunary POM ligand. The resulting complex has been utilized as a catalyst for the catalytic epoxidation of alkenes [32].

Compared to Ti-substituted POMs, the exploration of Zr-containing POMs has seldom been studied. In 1985, Chauveau et al. reported a compound of Zr-containing POM, [ZrW_5_O_19_H_2_]^2−^, which was considered as the Lindqvist-type structure. However, it is still doubted about the exact structure given the presented low signal-to-noise ratio and incorrect intensity ratio of ^183^W NMR data [33]. Subsequently, Villanneau et al. has been unambiguously determined the structure as [W_5_O_18_Zr(H_2_O)_3_]^2−^ by using EXFAS data [34,35]. Meanwhile, the same group also reported a similar compound with the structural formula of [{W_5_O_18_Zr(*µ*-OH)}_2_]^6−^ (Figure 2a). In 2009, Sokolov et al. reported two mono-Zr-substituted Dawson-type monomeric polyoxoanion clusters [{(H_2_O)_2_ZrP_2_W_17_O_61_}]^6−^ and [Zr(L-OOCCH(OH)CH_2_COO)P_2_W_17_O_61_]^8−^ (Figure 2b) through the reaction of mono-vacant α_2_-[P_2_W_17_O_61_]^10−^ ligand and Zr salt under aqueous solution conditions [36]. The latter complex exhibited chirality due to the presence of chiral L-malic acid ligand. Later on, other organic ligand-modified Zr-POM, (tpp)-Zr-(PW_11_O_39_)[TBA]_5_ (tpp referring to ternary porphyrin) (Figure 2c) [37] and (Pc)-Zr-(PW_11_O_39_)[TBA]_5_ (Pc referring to Phthalocyanine) [38], has been reported by Drain et al. in 2009 and 2013, respectively.

### 2.2. Ti/Zr-Substituted Dimeric POMs

In addition to the monomeric POMs, some presentative Ti/Zr-substituted dimeric POMs have also been reviewed herein in detail. In 1993, Finke et al. reported the first hexa-Ti-substituted sandwich-type dimeric silicotungstate, [A-*β*-Si_2_W_18_Ti_6_O_77_]^14−^ (Figure 3a), which was prepared by the reaction of [A-*β*-HSiW_9_O_34_]^9−^ with Ti(O)(C_2_O_4_)_2_^2−^ or Ti(O)SO_4_ in a regulated pH environment. The structural formulation has been lately corrected as [A-*β*-(SiW_9_O_37_)_2_(Ti-O-Ti)_3_]^14−^, implying the dimerization of two hypothetical “[A-*β*-SiW_9_(TiOH)_3_O_37_]^7−^” Keggin units through the linkage of Ti-O-Ti bridges [39]. In 2000, Kholdeeva and co-workers reported a Ti_2_-substituted dimeric POMs [(PTiW_11_O_39_)_2_OH]^7−^ which was prepared using [PTiW_11_O_40_]^5−^ subunit as reaction materials [25]. Then, Nomiya and co-workers prepared a similar hexa-Ti-substituted sandwich-type dimeric POMs except for the replacement of {A-SiW_9_O_34_} with {A-PW_9_O_34_} [40]. Subsequently, Cronin and co-workers reported a hexa-Ti-substituted tungstoarsenate, K_6_[Ti_4_(H_2_O)_10_(AsTiW_8_O_33_)_2_]·30H_2_O, where two {AsTiW_8_O_33_}fragments were used to encapsulate a {Ti_4_(H_2_O)_10_}^16+^ moiety [41]. Additionally, they also reported the first mono-Ti-substituted tungstoantimonate [TiO(SbW_9_O_33_)_2_]^16−^, which was two B*-α-*{Sb^III^W_9_O_33_} fragments linked by five sodium cations and an unprecedented square pyramidal Ti(O)O_4_ group with a terminal Ti = O bond. In 2013, Kortz et al. reported two Ti-substituted phosphotungstates, [Ti_8_(C_2_O_4_)_8_P_2_W_18_O_76_(H_2_O)_4_]^18−^ (Figure 3b) and [Ti_6_(C_2_O_4_)_4_P_4_W_32_O_124_]^20−^ (Figure 3c). The former is the Ti_8_-substituted Keggin-type phosphotungstates, consisting of two {PW_9_} units encapsulating eight titanium centers bridged with two Ti–O–Ti bonds. The latter represents the first Ti_6_-substituted Dawson-type phosphotungstates, which are constructed by two di-Ti-substituted {P_2_W_16_} units connected via two Ti(C_2_O_4_) moietes [42]. In 2015, Nomiya reported the first hexa-Ti-substituted Well–Dawson phosphotungstates [{*α*-P_2_W_15_Ti_3_O_60_(OH)_2_}_2_(Cp*Rh)_2_]^16−^ (Figure 3d). The polyoxoanion was constructed with two tri-Ti-substituted protonated Wells−Dawson subunits “[P_2_W_15_Ti_3_O_60_(OH)_2_]^10−^” bridged by the two organometallic Cp*Rh^2+^ groups [43]. Additionally, they also reported the first tetra-Ti^IV^-1,2-substituted *α*-Keggin polyoxotungstate in aqueous solution, [*α,α*-P_2_W_20_Ti_4_O_78_]^10−^ (Figure 3e). The polyoxoanion consisted of a dimeric anhydride form of two [α-1,2-PW_10_Ti_2_O_40_]^7−^ Keggin units linked with two Ti–O–Ti bonds [44]. Similar structures have also been reported by Mizuno and Wang’s groups, respectively [45,46]. With continuous research, more di-Ti-substituted POMs have been further developed. In 2007, Kortz et al. reported a special di-Ti-substituted tungstodiarsenate (III) [Ti_2_(OH)_2_As_2_W_19_O_67_(H_2_O)]^8−^ (Figure 3f), prepared by the reaction of TiOSO_4_ and K_14_[As_2_W_19_O_67_(H_2_O)] in a 2:1 molar ratio in acidic media (pH 2). The polyoxoanion was a sandwich-type structure with nominal *C_2v_* symmetry, which was constructed with two (B*-α-*As^III^W_9_O_33_) Keggin moieties linked by an octahedral {WO_5_(H_2_O)} fragment and two unprecedented square-pyramidal {TiO_4_(OH)} groups [47]. Nomiya et al. further reported the synthesis of a novel molecular solid Brønsted acid based on the Dawson-type sandwich POM [Ti_2_{P_2_W_15_O_54_(OH_2_)_2_}_2_]^8−^ (Figure 3g) [48]. Subsequently, they synthesized a similar structure using mono-lacunary Dawson precursor K_10_[*α*_2_-P_2_W_17_O_61_]·23H_2_O [49]. In 2015, Li’s group synthesized two isomorphic di-Ti-substituted Keggin-type phosphotungstate ([(Ti_2_O)(PW_11_O_39_)_2_]^8−^, Figure 3h) containing dissimilar copper under hydrothermal condition. The resulting organic–inorganic hybrid assemblies contained a rare corner-sharing double-Keggin type POM architecture in the Ti-POM species, which was further connected with the butterfly-type [Cu^II^Lo] units to form a 1-D chain and a square plane, respectively [50].

In contrast to the diverse structures of Ti-substituted POMs, Zr-substituted dimeric POMs have been far less reported. Some presentative examples are summarized below. In 2003, May et al. reported an example of a dimeric structure of mono-Zr-substituted Kegging-type POM, [Zr(PMo_12_O_40_)(PMo_11_O_39_)]^6−^ (Figure 4a), which also represented the first crystallographic determination of the [PMo_11_O_39_]^7−^ anion [51]. The similar Keggin-type chiral phosphotungstate and borotungstate were also reported by Liu and Xue’s groups in 2009, respectively [52,53]. In 2006, Nomiya et al. also reported the first Zr-substituted Well-Dawson phosphotungstate [Zr(*α*_2_-P_2_W_17_O_61_)_2_]^16−^ (Figure 4b), deriving from mono-lacunary precursor [*α*_2_-P_2_W_17_O_61_]^10−^ [54]. Similar mono-Zr-substituted Well-Dawson phosphotungstate POMs ([{P_2_W_15_O_54_(H_2_O)_2_}_2_Zr]^12−^, and [{P_2_W_15_O_54_(H_2_O)_2_}Zr{P_2_W_17_O_61_}]^14−^) have been further reported by Hill et al. in 2007 [55]. To increase the nuclearity of Zr centers, Kholdeeva et al. prepared three Zr_2_-substituted Keggin-type phosphotungstate ([{PW_11_O_39_Zr(*µ*-OH)}_2_]^8−^, [{PW_11_O_39_Zr(*µ*-OH)}_2_]^8−^, and [{PW_11_O_39_Zr}_2_(*µ*-OH)(*µ*-O)]^9−^) [56]. Then, Mizuno et al. synthesized a di-Zr-substituted Keggin-type silicotungstate [(*γ*-SiW_10_O_36_)_2_Zr_2_(*μ*-OH)_2_]^10−^ (Figure 4c) [57], and Sokolov et al. reported di-Zr-substituted Dawson-type phosphotungstate [{(H_2_O)Zr(*μ*_2_-OH)(P_2_W_17_O_61_)}_2_]^14−^ [58]. In 2011, Villanneau et al. reported two Zr_2_-containing POMs derivatives [{PW_9_O_34_{PO(R)}_2_}_2_{Zr(H_2_O)(*μ*-OH)}_2_]^4−^ and [{PW_9_O_34_{PO(R)}_2_}_2_{Zr(DMF)(*μ*-OH)}_2_]^4−^ (R = Ph, *^t^*Bu), which were obtained by using [(*^n^*Bu_4_N)_3_Na_2_[PW_9_O_34_{PO(R)}_2_] and ZrOCl_2_⋅8H_2_O [59]. Subsequently, the same group also reported a similar structure in 2013 [60]. In 2005, Hill et al. reported the first chiral tri-Zr-substituted Dawson-type phosphotungstate {[*α*-P_2_W_15_O_55_(H_2_O)]Zr_3_(*μ*_3_-O)(H_2_O)(L-tartH)[*α*-P_2_W_16_O_59_]}^15−^ (Figure 4d) and {[*α*-P_2_W_15_O_55_(H_2_O)]Zr_3_(*μ*_3_-O)(H_2_O)(D-tartH)[*α*-P_2_W_16_O_59_]}^15−^ [61]. After that, Cadot et. al. reported a tri-Zr(IV)-substituted sandwich-type Keggin POM [Zr_3_O(OH)_2_(SiW_9_O_34_)_2_]^12−^, which consists of a [Zr_3_O(OH)_2_] triangular central cluster closely embedded between two A-α-[SiW_9_O_34_]^10−^ subunits [62]. Subsequently, three cases of isomorphic compounds were also reported by Xue, Nomiya and Yang’s groups, respectively [63,64,65]. Among these tri-Zr-substituted POMs, it is worth mentioning that Yang’s group reported the first tri-Zr^IV^-substituted POM [Zr_3_(*µ*_2_-OH)_2_(*µ*_2_-O)(A-*α*-GeW_9_O_34_)(1,4,9-*α*-P_2_W_15_O_56_)]^14−^ (Figure 4e), where the tri-Zr centers were stabilized by mixed types of tri-lacunary POM ligands including Keggin-type [A-*α*-GeW_9_O_34_]^10−^ and Dawson-type [1,4,9-*α*-P_2_W_15_O_56_]^12−^ units [66]. In addition, a number of tetra-Zr substituted POMs have also been prepared. For instance, Pope et al. reported a Zr_4_-substituted phosphotungstate, [Zr_4_(*µ*_3_-O)_2_(*µ*_2_-OH)_2_(H_2_O)_4_(P_2_W_16_O_59_)_2_]^14−^ (Figure 4f). Therein, the divacant lacunary {P_2_W_16_O_59_} ligands were derived from the plenary Wells–Dawson (*α*-P_2_W_18_O_62_) polyoxoanion [67]. Then, similar structures were reported by Hill and Li’s groups in 2005 and 2013, respectively [68,69]. Subsequently, tetra-Zr substituted Keggin-type silicotungstates [(*γ*-SiW_10_O_36_)_2_Zr_4_(*µ*_4_-O)(*µ*-OH)_6_]^8−^ (Figure 4g) and five other similar structures were successively reported [70,71,72,73,74,75,76,77]. The nuclearity of Zr substitution has been further improved to six by Kortz and co-workers. they reported the first hexa-Zr-substituted dimeric tungstoarsenates [Zr_6_O_4_(OH)_4_(H_2_O)_2_(CH_3_COO)_5_(AsW_9_O_33_)_2_]^11−^ (Figure 4h). In the polyoxoanion, the unprecedented hexa-Zr unit is perfectly located at the cavity formed by two (B-*α*-AsW_9_O_33_) fragments lying at an angle of about 74° with respect to each other [78].

### 2.3. Ti/Zr-Substituted Trimeric POMs

In contrast to the dimeric POM structures, the syntheses of Ti/Zr-substituted trimeric POMs have rarely been reported. The early reported Zr-substituted trimeric POM is Zr_6_O_2_(OH)_4_(H_2_O)_3_(*β*-SiW_10_O_37_)_3_]^14−^ (Figure 5a), which was synthesized using an aqueous solution-based method by Kortz et al. in 2006. This polyoxoanion consists of three *β*_23_-SiW_10_O_37_ units linked by an unprecedented Zr_6_O_2_(OH)_4_(H_2_O)_3_ cluster with *C_1_* point group symmetry [79]. The similar polyoxoanion [Zr_6_(*μ*_3_-O)_3_(OH)_3_(OAc)(H_2_O)(*β*-GeW_10_O_37_)_3_]^16−^ was also synthesized via hydrothermal method by Yang’s group in 2019, except that {*β*-SiW_10_O_37_} was replaced by {*β*-GeW_10_O_37_} [80]. Late on, Kortz et al. reported another Zr_6_-substituted silicotungstate [Zr_6_(O_2_)_6_(OH)_6_(*γ*-SiW_10_O_36_)_3_]^18−^ (Figure 5b) in 2008, where 6-Peroxo-6-Zr Crown embedded in a triangular polyoxoanion. This polyoxoanion is composed of three [*γ*-SiW_10_O_36_]^8−^ units encapsulating the unprecedented [Zr_6_(O_2_)_6_(OH)_6_]^6+^ wheel, while it can also be considered as a cyclic assembly of three fused {Zr_2_(O_2_)_2_(OH)_2_(*γ*-SiW_10_O_36_)} monomers. This work belongs to the first structurally characterized Zr-peroxo POM with side-on, bridging peroxo units [81]. Additionally, Kortz et al. reported the first examples of Ti-containing trimeric polytungstates, two cyclic Ti_9_-containing trimeric POMs, [(*α*-Ti_3_PW_9_O_38_)_3_(PO_4_)]^18−^ (Figure 5c) and [(*α*-Ti_3_SiW_9_O_37_OH)_3_(TiO_3_(OH_2_)_3_)]^17−^ (Figure 5d) using the solution-based synthetic method. Both compounds were composed of three (Ti_3_XW_9_O_37_) units (X = P or Si) bridged with three Ti-O-Ti bonds and a capping group (tetrahedral PO_4_ or octahedral TiO_6_) [82]. Liu’s group reported two similar trimeric nine-Ti^IV^ contained tungstogermanates {K⸦[(Ge(OH)O_3_)(GeW_9_Ti_3_O_38_H_2_)_3_]}^14−^ (Figure 5e) and {K⸦[(SO_4_)(GeW_9_Ti_3_O_38_H_3_)_3_]^10−^. The two compounds were obtained from the reactions between K_8_[*γ*-GeW_10_O_36_] [83] and TiO(SO_4_) under different pH conditions. The former polyoxoanion consisted of three tri-Ti^IV^-substituted Keggin fragments [GeW_9_Ti_3_O_38_] and a GeO_4_ tetrahedral linker bridged with both Ti–O–Ti and Ti–O–Ge bonds, and the structure of the latter one is similar except for the replacement of SO_4_ with GeO_4_ [84]. Recently, a novel trimer compound [{Ca_6_(CO_3_)(*μ*_3_-OH)(OH_2_)_18_}(P_2_W_15_Ti_3_O_61_)_3_Ca(OH_2_)_3_]^19−^ (Figure 5f) was reported that contains a hexacalcium cluster cation, one carbonate anion, and one calcium cation assembled on a trimeric tri-Ti-substituted Wells–Dawson polyoxometalates. This complex was obtained through the reaction of calcium chloride with the monomeric trititanium(IV)-substituted Wells−Dawson POM species “[P_2_W_15_Ti_3_O_59_(OH)_3_]^9−^”. During the synthesis, the [Ca_6_(CO_3_)(*μ*_3_-OH)(OH_2_)_18_]^9+^ cluster cation, composed of six calcium cations linked by one *μ_6_*-carbonato anion and one *μ_3_*-OH^-^ anion, assembled with one calcium ion, a trimeric “[P_2_W_15_Ti_3_O_59_(OH)_3_]^9−^” species to form the target product. The compound is an unprecedented POM species containing an alkaline-earth-metal cluster cation, and it is also the first example of alkaline-earth-metal ions clustered around a Ti-substituted POM [85].

### 2.4. Ti/Zr-Substituted Tetrameric POMs

In this section, a number of Ti/Zr-substituted tetrameric POMs will be briefly introduced. The early example is a dodeca-Ti-substituted Dawson-type tetrameric, [{Ti_3_P_2_W_15_O_57.5_(OH)_3_}_4_]^24−^ (Figure 6a), representing a supramolecular phosphotungstate reported by Kortz’s group in 2003 [86]. The polyoxoanion was composed of four lacunary [P_2_W_15_O_56_]^12−^ Well–Dawson building blocks linked with terminal Ti-O bonds, resulting in a structure with *T_d_* symmetry. The {Ti_12_O_46_} core of the polyoxoanion is composed of four groups of three edge-shared, corner-linked TiO_6_ octahedra. Such a rare arrangement resembles one set of the four corner-shared faces of an octahedron, described as a “reversed Keggin structure”, which is very similar to the [As_4_Mo_12_O_50_]^8−^ geometry reported by Sasaki and Nishikawa [87]. Apart from this Ti_12_ cluster, they also discovered another deca-Ti-substituted tetrameric species [{Ti_3_P_2_W_15_O_57.5_(OH)_3_}_2_{Ti_2_P_2_W_16_O_60_(OH)}_2_]^26−^ containing two {Ti_3_P_2_W_15_} and two {Ti_2_P_2_W_16_} fragments, therefore resulting in a structure with *C_2v_* symmetry. Meanwhile, Nomiya et al. also reported two multi-Ti-substituted tetrameric POMs. The first one is a giant “tetrapod”-shaped dodeca-Ti-substituted Dawson-type tetrameric phosphotungstate, [(*α*-1,2,3-P_2_W_15_Ti_3_O_60.5_)_4_Cl]^37−^ (Figure 6b), which contains the four Wells-Dawson units fused together through Ti–O–Ti bonds. The structure exhibits an approximately *T_d_* symmetry, where the four Ti_3_O_6_ facets of “P_2_W_15_Ti_3_” occupied four alternate facets of an octahedron, and the one Cl^-^ ion was encapsulated in the central octahedral cavity [88]. It is noted that a similar structure [(P_2_W_15_Ti_3_O_60.5_)_4_(NH_4_)]^35−^ was also reported in 2011, except that Cl^-^ was replaced by NH_4_^+^ [89]. The other example belongs to a “tetrapod”-shaped Ti_16_-substituted Dawson-type tetrameric phosphotungstate [(*α*-1,2,3-P_2_W_15_Ti_3_O_62_)_4_{*μ*_3_-Ti(OH)_3_}_4_Cl]^45−^ (Figure 6c). The polyoxoanion, prepared by the reaction of [P_2_W_15_O_56_]^12−^ with an excess amount of TiCl_4_ in aqueous solution, was composed of four tri-Ti^IV^-1,2,3-substituted α-Dawson substructures, four Ti(OH)_3_ bridging groups, and one encapsulated chloride ion [90]. Subsequently, Nomiya et al. also reported three similar Ti_16_-substituted POMs, [(*α*-P_2_W_15_Ti_3_O_59_(OH)_3_)_4_{*μ*_3_-Ti(H_2_O)_3_}_4_X]^21−^ (X = Br^−^, I^−^, and NO_3_^−^), except that halogen atoms and some oxygen atoms are protonated [91]. In 2004, Kortz et al. synthesized a unique cyclic octa-Ti- substituted tetrameric tungstosilicate [{*β*-Ti_2_SiW_10_O_39_}_4_]^24−^ (Figure 6d) assembly under mild, one-pot reaction conditions. The polyoxoanion is composed of four {*β*-Ti_2_SiW_10_O_39_} Keggin fragments bridged with Ti–O–Ti bonds, leading to a cyclic assembly. The successful preparation of this compound provides future possibilities for preparing even larger wheel-shaped polyoxotungstates and other discrete nanomolecular objects of similar size, structure, and function as those made with polyoxometalates [92]. Then, Kortz’s group also successfully prepared a hepta-Ti substituted arsenotungstate [Ti_6_(TiO_6_)(AsW_9_O_33_)_4_]^20−^ (Figure 6e) in 2014 using a simple one-pot procedure. The polyoxoanion contains a novel Ti_7_-core consisting of a central TiO_6_ octahedron surrounded by six TiO_5_ square pyramids, which was further capped by four trilacunary {As^III^W_9_} fragments, leading to an assembly with *T_d_* point-group symmetry [93]. In 2019, Yang’s group also reported a similar structure [Ti_7_O_6_(SbW_9_O_33_)_4_]^20−^ by replacing {AsW_9_O_33_} with {SbW_9_O_33_} [94]. In 2022, Yang’s group further synthesized a ring-shaped 12-Ti-substituted poly(polyoxometalate) [{K_2_Na(H_2_O)_3_}@{(Ti_2_O)_2_(Ti_4_O_4_)_2_(A-*α*-1,3,5-GeW_9_O_36_)_2_(A-*α*-2,3,4-GeW_9_O_36_)_2_}]^25−^ (Figure 6f) under hydrothermal conditions, which represents the highest number of Ti centers in Keggin-type poly(POM) family to date. In this structure, two types of novel chiral trivacant [GeW_9_O_36_]^14−^ (A-*α*-1,3,5-GeW_9_O_36_ and A-*α*-2,3,4-GeW_9_O_36_) fragments have been first discovered in POM chemistry, and four [GeW_9_O_36_]^14−^ fragments are alternately connected by two Ti_2_O and two Ti_4_O_4_ cores to form a ring-shaped poly(POM) [95].

In addition to these Ti-substituted POMs, some Zr-substituted tetrameric POMs have also been actively investigated. For instance, a Zr_4_-substituted tungstoselenites, [(*α*-SeW_9_O_34_){Zr(H_2_O)}{WO(H_2_O)}(WO_2_)(SeO_3_){*α*-SeW_8_O_31_Zr(H_2_O)}]_2_^12−^ (Figure 7a), has been firstly constructed using {*α*-SeW_9_} building blocks. Such a tetrameric structure can be divided into two same subunits, each containing a dimer sandwich-type structure that consists of a well-known trivacant Keggin-type {*α*-SeW_9_O_34_} building blocks [96]. Then, Yang’s group continuously reported eight tetrameric Zr-substituted POMs. The first example represents a new tetra-Zr-substituted tungstophosphate, {Zr_2_[SbP_2_W_4_(OH)_2_O_21_][*α*_2_-PW_10_O_38_]}_2_^20−^ (Figure 7b) through the hydrothermal reaction of the [B-*α*-SbW_9_O_33_]^9−^ building block with Zr^4+^ cations and PO_4_^3−^ anions in the presence of dimethylamine hydrochloride in NaOAc-HOAc buffer solution. The compound exhibits a toroidal structure formed by two divacant [*α*_2_-PW_10_O_38_]^11−^ units and two [SbP_2_W_4_(OH)_2_O_21_]^7−^ fragments linked by four Zr^4+^ cations. It is noted that the triangular pyramidal SbO_3_ in the [B-α-SbW_9_O_33_]^9−^ precursor was replaced by tetrahedral PO_4_ unit in the final compound, and the pendant SbO_3_ derives from the dissociation of the [B-*α*-SbW_9_O_33_]^9−^ precursor [97]. Then, the same group also reported a Zr(IV)-substituted tetramer polyoxotungstate, [Zr_4_(*β*-GeW_10_O_38_)_2_(A-*α*-PW_9_O_34_)_2_]^26−^ (Figure 7c) [98], which was obtained in a one-pot reaction of the hexalacunary polyanion [P_2_W_12_O_48_]^12−^ and trilacunary [GeW_9_O_34_]^10−^ polyanion precursors with Zr^4+^ in a slightly alkali aqueous solution in the presence of borates. Subsequently, a Zr_9_-substituted tetrameric germanotungstate, [{Zr_5_(*μ*_3_-OH)_4_(OH)_2_}@{Zr_2_(OAc)_2_(*α*-GeW_10_O_38_)_2_}_2_]^22−^ (Figure 7d) was constructed by two novel sandwich-type dimers [Zr_2_(OAc)_2_(*α*_2_-GeW_10_O_38_)_2_]^18−^ and one unique [Zr_5_(*μ*_3_-OH)_4_(OH)_2_]^14+^ core in an approximately orthogonal fashion, showing a staggering tetrahedral polyoxoanion [80]. In addition, Yang et al. also reported a series of ring-shaped Zr_8_-substituted silicotungstates, [{Zr_2_(OH)_2_(*α*-SiW_10_O_38_)}_2_{Zr_2_(OH)_2_(*β*-SiW_10_O_38_)}_2_]^24−^ (Figure 7e) (dap = 1,3-diamino-propane), [(Zr_2_(OH)_2_)_2_(Zr_2_BO(OH)_4_)_2_(*β*-SiW_10_O_38_)_4_]^26−^ (Figure 7f) and [(Zr_2_BO(OH)_4_)_2_(Zr_2_B_2_O_2_(OH)_5_)_2_(*β*-SiW_10_O_38_)_4_]^28−^ [99]. The latter two compounds first provided the possibility of introducing Zr–B–O linkage to the lacunary sites. Very recently, they also reported a di-Zr-substituted polyoxotungstate [ZrSb_4_(OH)O_2_(A-*α*-PW_8_O_32_)(A-*α*-PW_9_O_34_)]_2_^18−^ (Figure 7g) [100] and hepta-Zr-incorporated polyoxometalate [SbZr_7_O_6_(OH)_4_(B-*α*-GeW_9_O_34_)_2_(B-*α*-GeW_11_O_39_)_2_]^21−^ (Figure 7h) [101], these works greatly enriched the family of Zr-substituted POMs.

### 2.5. Ti/Zr-Substituted Multimeric POMs

Compared to those mono-, di-, tri- and tetrameric POM structures, there are very few reports on the preparation of multimeric Ti/Zr-substituted POMs. To date, only two related compounds have been reported. The first example belongs to an octa-Ti-substituted Dawson-type supramolecular polyoxoanion reported by Kortz’s group in 2003 [86]. However, the polyoxoanion was described by the preliminary and incomplete formula “Ti_8_P_12_W_84_” or “(Ti_2_P_2_W_15_)_2_(Ti_2_P_2_W_16_)_2_(P_2_W_11_)_2_” (Figure 8a) due to the poor quality of the crystallographic data. The second representative example is the gigantic Zr_24_-cluster-substituted Keggin-type germanotungstates [Zr_24_O_22_(OH)_10_(H_2_O)_2_(W_2_O_10_H)_2_(GeW_9_O_34_)_4_(GeW_8_O_31_)_2_]^32−^ (Figure 8b) reported by Yang’s group in 2014 [102]. The polyoxoanion was successfully synthesized under hydrothermal conditions, which contains the largest [Zr_24_O_22_(OH)_10_(H_2_O)_2_] cluster among all reported Zr-based poly(polyoxometalate)s to date. Detailed structural analyses of this complex showed that the centrosymmetric Zr_24_-cluster-based hexamer contained two symmetry-related [Zr_12_O_11_(OH)_5_(H_2_O)(W_2_O_10_H)(GeW_9_O_34_)_2_(GeW_8_O_31_)]^16−^ trimers linked via six *μ*_3_-oxo bridges, which were further encapsulated by different POM fragments including B-*α*-GeW_9_O_34_, B-*α*-GeW_8_O_31_, and W_2_O_10_. Catalytic experiments also showed that this compound worked as a good catalyst for the oxygenation of thioethers to sulfoxides/sulfones in the presence of H_2_O_2_, which could be attributed to the unique redox property of oxygen-enriched polyoxotungstate fragments as well as the Lewis acidity of the Zr_24_ cluster. Although the synthesis of multimeric POMs is rather difficult, these pioneering works provide some insights and future direction for the exploration of this specific research area.

## 3. The Applications of Representative Ti/Zr-Substituted POMs

It is well known that transition-metal clusters are a unique area in inorganic chemistry, considering their vital contribution to the blossom of modern chemistry as well as their potential application as structural models for various industrial and biological catalytic processes [103,104,105,106]. To date, the Ti/Zr-substituted POMs have been widely investigated as catalysts for the oxidation of organic substrates. For example, Poblet et al. investigated the oxidation of alkenes by H_2_O_2_ catalyzed using Ti(IV)-containing POMs, which were models of Ti single-site catalysts at the DFT computational level. The catalytic mechanism of the C_2_H_4_ epoxidation with H_2_O_2_ mediated by [PTi(OH)W_11_O_39_]^4−^ and [Ti_2_(OH)_2_As_2_W_19_O_67_(H_2_O)]^8−^ can be processed by following two main steps (Figure 1): (i) H_2_O_2_ was activated to form the titanium-peroxo or -hydroperxo intermediate, and (ii) the reactive intermediate further attack alkene to form the epoxide and water [107]. Kholdeeva and co-workers also reported several cases of various organic catalytic studies with Ti-containing POMs. For example, they investigated the mechanism of thioether oxidation of (Bu_4_N)_7_{[PW_11_O_39_Ti]_2_OH} dimeric heteropolytungstate in 2000 [108]. In 2004, they also reported a protonated titanium peroxo complex [Bu_4_N]_4_[HPTi(O_2_)W_11_O_39_] and found that protonated titanium peroxo complex has a higher redox potential, which can improve the catalytic performance [109]. In the same year, Ti(IV)-monosubstituted Keggin-type POMs were reported to exhibit excellent catalytic oxidation properties with H_2_O_2_ [110,111,112]. Subsequently, Kholdeeva et al. also reported the epoxidation of a range of alkenes easily proceeds with aqueous H_2_O_2_ as oxidant and the dititanium-containing 19-tungstodiarsenate (III) as catalyst [113]. In 2012, the same group also published two works on alkene oxidation by Ti-containing POMs. They claimed that the energy barrier for the heterolytic oxygen transfer from the reactive Ti hydroperoxo intermediate was significantly reduced by the protonated Ti-containing POM, as revealed by the kinetic and DFT studies, thereby greatly enhancing the activity and selectivity of alkene oxidation [114,115]. Subsequently, Poblet and Guillemot reported the alkene epoxidation catalyzed by the hybrid [B-*α*-SbW_9_O_33_(*^t^*BuSiO)_3_Ti(O*^i^*Pr)]^3−^, [PW_9_O_34_(*^t^*BuSiO)_3_Ti(O*^i^*Pr)]^3−^, and Ti-complexe of silanol functionalized POMs [SbW_9_O_33_(RSiO)_3_Ti(O*^i^*Pr)]^3−^, respectively [32,116,117]. Based on the research of alkene epoxidation catalysis, Kholdeeva et al. also revealed the mechanism of thioether oxidation of Ti-substituted POMs by kinetic modeling and DFT calculations. Two possible models regarding the active group were proposed: (1) the active group is the terminal Ti−OH group for the mononuclear, and (2) the active group is the bridging Ti_2_(*μ*-OH) moiety for the multinuclear [118]. Subsequently, Yang’s group also reported two cases of the catalytic oxidation of thioethers using Ti_7_- and Ti_12_-substituted POMs, respectively, which both exhibited good catalytic properties [94,95]. In addition, Li’s group investigated the photocatalytic degradation of MB with Ti_2_-substituted POM units under UV irradiation, which proved that Ti-substituted Keggin-type POMs showed better photocatalytic activities than that of typical Keggin-type POMs [50]. Some Ti-substituted POMs have also been reported with good electrocatalytic properties [119].

Compared with the applications of Ti-substituted POMs, a part of Zr-substituted POMs also exhibited good catalytic oxidation of thioether, electrocatalytic and nonlinear optical properties, etc. For example, Yang’s groups reported a few works on the oxidation of sulfide using Zr_2_-, Zr_4_-, Zr_7_- and Zr_8_-substituted POMs, respectively [76,99,100,101]. These compounds showed remarkable heterogeneous catalysts for the catalytic oxidation of sulfides into the corresponding sulfones with H_2_O_2_. The same groups also reported several Chiral Zr-substituted POMs which have excellent nonlinear optical properties [75,77]. In addition, transition-metal-substituted POMs are air- and water-stable Lewis acids that were often used in organic reactions [120], and in the last two decades, most Zr-substituted POMs also were reported to be used as optimal Lewis acid catalysts to hydrolyze the O = C-NH- bonds in proteins or peptides (Figure 2), leading to the formation of amino acids [121,122,123,124,125,126]. These works implied the potential applications of Zr-substituted POMs in biological systems.

## 4. Conclusions and Perspectives

In summary, this review has mainly addressed the development of Ti/Zr-substituted POMs with an emphasis on structural diversity, synthetic approaches, and potential catalytic applications. According to the overview of these reported Ti/Zr-substituted POMs, we can conclude that (1) the solution-based synthetic approach is an effective method for the syntheses of Ti/Zr-substituted POMs; (2) interesting and attracting POM structures could be often obtained via the hydrothermal synthetic strategy. These successful synthetic strategies and the persistent and dedicated efforts of chemists over the past decades have greatly contributed to the vast and beautiful array of Ti/Zr-substituted POMs. However, there are still bottlenecks in Ti/Zr-substituted POM chemistry with respect to the controllable assembly of target POM structures, the exploration of novel synthetic approaches, as well as the insightful understanding of catalytic mechanisms using Ti/Zr-substituted POM catalysts. Therefore, we believe that the development of other synthetic methods, for instance, the mixed solvent diffusion method, ionothermal approach, templated modular assembly method or the combination with existing solution/hydrothermal approaches, would provide new blood to the synthetic chemistry of Ti/Zr-substituted POMs. Moreover, the ground-breaking exploration of new catalytic functionalities of these Ti/Zr-substituted POMs should also be strengthened in the future. Finally, we hope this critical review could provide research insights into the controllable design and syntheses of Ti/Zr-substituted POMs derivatives and, in the meantime, attract more researchers to join the research community of POM Chemistry or the related interdisciplinary research areas.

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
