# Peer review of "Recent Advances of Ti/Zr-Substituted Polyoxometalates: From Structural Diversity to Functional Applications"

_molecules, 2022, doi:10.3390/molecules27248799_

Round 1
Reviewer 1 Report
This review manuscript by Ni and co-authors has summarized recent advances of Ti/Zr-substituted polyoxometalates. The authors provided a comprehensive overview of the structural variations of Ti/Zr-substituted POMs, which classified by the polymerization number of POM building blocks, including monomers, dimers, trimers, tetramers and multimers. In addition, the potential applications of Ti/Zr-substituted POMs have also been briefly introduced, such as oxidation of organic substrates and hydrolysis of proteins or peptides. Overall, this review manuscript is well-organized and quite comprehensive, which could attract a broad readership of Molecules. Therefore, I would recommend its acceptance after revising the following problems.
1. Are all the Figures obtained copyright permissions from the publisher? Please note it in the captions of all Figures if applicable.
2. Some Figures, such as Figures 1e, 2a, 3d, 6b, are not clear enough, replace these pictures with high-resolution ones?
3. The authors should carefully check and revise the format of some references, since I found many mistakes after a short look. For example, the title of Ref. 70 and 83, please revise it.
4. On page 4, line 4 in the text, the molecular formula is incorrectly written. for example: “ [(H2O)4(3-Hpic)2-Ln][(H2O)5(3-Hpic)2Ln][PW10Ti2O40]-”. On page 14, line 20 in the text, the molecular formula is incorrectly written. for example: “[Zr12O11(OH)5(H2O)(W2O10H)(GeW9-O34)2(GeW8O31)]16-”.
Author Response
Dear Reviewer,
Thank you very much for your comments. Attached please find our point-by-point responses to your comments.
Best regards,
Hongjin

Reviewer 2 Report
Comments can be found in the attached file

Author Response
Dear Reviewer,
Thank you very much for your valuable comments. Attached please find our responses to the comments.
Best regards,
Hongjin

Reviewer 3 Report
The present review aims at providing an extensive compilation of Ti and Zr derivatives of polyoxometalates and is mainly focused on the structural determination of the complexes described in the literature. This structural part represents the very large part of the manuscript, which shows the richness of this chemistry within the polyoxometalate community.
However, some references are missing, especially concerning the works of Kholdeeva and co-workers who is a pioneer in this chemistry (first structural publications in the early 2000ies: see the following references for Ti4+ (Inorg. Chem. 2000, 39, 3828-3837) and Zr4+ complexes (Inorg. Chem. 2006, 45, 7224-7234). The work of the Paris group deserved also to be mentioned, concerning the use of hybrid derivatives of POMs for their coordination chemistry (see the references, Inorg. Chem. 2011, 50, 1164–1166 and Eur. J. Inorg. Chem. 2013, 1815–1820).
Furthermore, the following sentence concerning the exact structure of the [W5O18Zr(H2O)]2- is incorrect: “However, the weakness of the presented 183W NMR data (low signal-to-noise ratio and incorrect intensity ratio) has caused some casting doubt on the real structure of [ZrW5O19H2]2- polyoxoanion”. Indeed the exact structure has been unambiguously determined using EXFAS data and showed to be [W5O18Zr(H2O)3]2- as mentioned in the following missing reference: Inorg. Chem. 2006, 45, 1915-1923. Please modified the text according to the conclusions of these two papers.
Most important, the part 3 of the manuscript, which deals with the utilization of Ti and Zr derivatives of POMs, is incomprehensibly small compared to the structural description of the complexes. However, this is the richness and main interest of this chemistry, especially for oxidation reactions which have been studied in great detail by Kholdeeva and co-workers (see the references 101-103 and the following missing references which should be added: J. Mol. Catal. A: Chemical 158 2000. 223–229, Inorg. Chem. 2004, 43, 2284-2292, Inorg. Chem. 2005, 44, 1635-1642, J. Mol. Catal. A: Chemical 232 (2005) 173–178, J. Mol. Catal. A: Chemical 262 (2007) 7–24, Eur. J. Inorg. Chem. 2010, 5312–5317) and Guillemot and Poblet (reference 33 and the missing references which should be added: ACS Catal. 2018, 8, 2330−2342, ChemCatChem 2021, 13, 1220–1229). This section needs to be extensively rewritten and much more detailed in view of the catalytic performance of these compounds and the number of meticulous studies published over the last 20 years. There is a lot to be said for the reactivity of Ti4+ complexes and the formation of their peroxo complexes. The use of Zr POMs as Lewis acid catalysts for various organic reaction should also be mentioned (see the following reference: Chem. Eur. J. 2010, 16, 7256 – 7264).
In conclusion, to my opinion the present work is of great interest but should be modified and completed following the above comments and requests. I recommend publication of this manuscript with minor revision, providing that these changes have been added to the text.
Author Response

(The authors gave the same response as above.)
